# Concordance Rate of Colposcopy in Detecting Cervical Intraepithelial Lesions

**DOI:** 10.3390/diagnostics12102436

**Published:** 2022-10-08

**Authors:** Frederik A. Stuebs, Anna K. Dietl, Annika Behrens, Werner Adler, Carol Geppert, Arndt Hartmann, Antje Knöll, Matthias W. Beckmann, Grit Mehlhorn, Carla E. Schulmeyer, Paul Gass, Martin C. Koch

**Affiliations:** 1Department of Gynecology and Obstetrics, Erlangen University Hospital, Comprehensive Cancer Center Erlangen—European Metropolitan Area of Nuremberg (CCC ER-EMN), Friedrich Alexander University of Erlangen—Nuremberg, Universitaetsstrasse 21–23, 91054 Erlangen, Germany; 2Department of Medical Informatics, Biometry and Epidemiology, Friedrich Alexander University of Erlangen—Nuremberg, Waldstrasse 6, 91054 Erlangen, Germany; 3Institute of Pathology, Erlangen University Hospital, Comprehensive Cancer Center Erlangen—European Metropolitan Area of Nuremberg (CCC ER-EMN), Friedrich Alexander University of Erlangen—Nuremberg, Krankenhausstrasse 8–10, 91054 Erlangen, Germany; 4Institute of Clinical and Molecular Virology, Erlangen University Hospital, Friedrich Alexander University of Erlangen—Nuremberg, Schlossgarten 4, 91054 Erlangen, Germany; 5Gynecology Consultancy Practice, German Cancer Society [DKG] and Committee on Cervical Pathology and Colposcopy [AG-CPC] Certified Gynaecological Dysplasia Consultancy Practice, Frauenarztpraxis Erlangen, Neustädter Kirchenplatz 1a, 91054 Erlangen, Germany; 6Department of Gynecology and Obstetrics, Hospital ANregiomed Ansbach, Escherichstrasse 1, 91522 Ansbach, Germany

**Keywords:** cervical intraepithelial neoplasia, colposcopy, cervical cancer, cervical cancer screening assessment

## Abstract

Background: The purpose of this research is to estimate the rate of concordance, sensitivity, specificity, positive predictive value (PPV) and negative predictive value (NPV) of colposcopy for high-grade squamous lesions and carcinomas (HSIL+). Methods: We conducted a retrospective study of colposcopies performed in the certified Dysplasia Unit in Erlangen between January 2015 and May 2022 (7.5 years). The colposcopic findings were correlated with biopsies obtained during examinations or surgery. Cases without histology were excluded. The primary outcome was the rate of concordance between the colposcopic and histological findings in relation to the type of transformation zone (TZ), examiner’s level of experience and age of the patients. Results: A total of 4778 colposcopies in 4001 women were analyzed. The rates of concordance for CIN I/LSIL, CIN II/HSIL, CIN III/HSIL, and carcinoma were 43.4%, 59.5%, 78.5%, and 53.9%, respectively. The rate of concordance was lowest for TZ3 and highest for colposcopists with more than 10 years’ experience. Conclusions: Colposcopy is an important, feasible, and effective method. Careful work-up needs to be performed for women with TZ3 who are over 35 years old, as they are at the highest risk of being misdiagnosed. The highest concordance for detecting HSIL+ was seen for colposcopists with >10 years’ experience.

## 1. Introduction

Cervical cancer is one of the most common cancers amongst women worldwide [1,2,3,4]. In 2018, there were approximately 570,000 cases of cervical cancer and 311,000 deaths worldwide, and in Germany the incidence was 4320 women in 2018, 1612 of whom died of cervical cancer [1,5]. There are geographic disparities throughout the world. In recent decades, the incidence of cervical cancer has declined in developed countries, mainly due to the effects of nationwide screening programs [6]. High-grade squamous lesions (HSIL) and cervical cancers are caused by persistent infection with human papillomavirus (HPV) [7]. The detection of HSIL plays a crucial role, as HSIL can progress to cervical cancer [8]. Suspicious cytological findings obtained in screening programs are, in many cases, among the first signs of HSIL or invasive cancer; women with abnormal cytology are referred to certified dysplasia units or gynecological centers [6,9]. These dysplasia units and gynecological centers are certified by the German Cancer Society (*Deutsche Krebsgesellschaft e. V.* [DKG]), the Working Group for Gynecological Oncology (*Arbeitsgemeinschaft Gynäkologische Onkologie e. V.* [AGO]), the Working Group on Cervical Pathology and Colposcopy (*Arbeitsgemeinschaft Zervixpathologie & Kolposkopie* [AG-CPC]), and the German Society for Gynecology and Obstetrics (*Deutsche Gesellschaft für Gynäkologie und Geburtshilfe e. V.* [DGGG]) [9].

Cytology is the most accessible, feasible and cost-effective tool for the screening of HSIL and invasive cancer [10,11]. There is considerable debate regarding the accuracy of Pap smears, which is reported to range between 53% and 78% [10]. The combination of colposcopy, cytology, and colposcopy-directed biopsy is the gold standard for diagnosing HSIL and cervical cancer [6,12,13]. Although cervical cancer screening is highly effective, the management of HSIL is a public health challenge [14]. 

Colposcopy is the most basic first-step and cost-effective examination for accurate diagnosis of HSIL in women who are referred due to abnormal cytology [15]. Colposcopy enables the examiner to localize potential lesions, evaluate the lesion’s severity, and obtain a colposcopically directed biopsy [15]. Colposcopy is reported to have the best efficacy in detecting HSIL compared with conventional Pap smears and liquid-based Pap smears [16]. Colposcopy is better at differentiating between HSIL and low-grade squamous lesions (LSIL) than differentiating LSIL from a normal cervix [17]. Nevertheless, colposcopy remains a subject of clinical interest [15]. Wide ranges of sensitivity and specificity rates are reported in the detection of HSIL, at 30–90% and 44–97%, respectively [15,16,17]. A total of 30% of HSILs are reported to be missed by colposcopy alone [18]. In order not to overlook HSIL or cervical cancer, a colposcopy-directed biopsy is necessary, which is an invasive procedure that is associated with certain complications, including pain, bleeding, and fibrosis [10]. It is also important to avoid overtreatment of HSIL, such as large loop excision of the transformation zone (LLETZ) or loop electrosurgical excision procedures (LEEP) with laser coagulation of the periphery. The morbidity associated with the procedure, such as perinatal mortality (relative risk 2.87), preterm delivery (<32/34 weeks; relative risk 2.59–2.78), or low birth weight (<2000 g; relative risk 2.53–2.86) needs to be avoided [6,19,20,21]. 

There are possible explanations for the wide range of sensitivity and specificity rates for colposcopy, such as the level of experience of the colposcopists, the type of transformation zone and the age of the patients. The aim of the present study was to determine the concordance of colposcopy in comparison with the final histological diagnosis obtained with an LLETZ, LEEP or laser conization.

## 2. Materials and Methods

Between January 2015 and May 2022, 11,086 colposcopies of the cervix were performed in the nationally certified dysplasia unit at Erlangen University Hospital (Figure 1). Abnormal cervical cytology was the most common reason for women being referred to the dysplasia unit. All patients (*n* = 4778) who had a biopsy or underwent excisional surgical treatment—LLETZ, loop electrosurgical excision procedure (LEEP) with laser coagulation of the periphery or laser conization—were included. Patients in whom ablation was performed without a biopsy being taken during the colposcopic examination were excluded from the analysis due to the lack of histological findings. Only women with an adequate colposcopy were included.

In our department, colposcopies are performed in standardized conditions using a Zeiss KSK 150 FC colposcope. The general assessment was carried out in accordance with the 2011 International Federation for Cervical Pathology and Colposcopy (IFCPC) colposcopic terminology for the cervix: “adequate” or “inadequate” for the reason (e.g., inflammation, bleeding, scar); squamocolumnar junction visibility (completely visible, partially visible, not visible); and transformation zone (TZ) types 1, 2, or 3 [22,23,24]. A conventional Pap smear of the cervix, a test for human papillomavirus (hybrid capture test 2, 2015–2018; Abbott RealTime high-risk HPV assay on an Abbott m2000sp, 2019–2020; or Roche, cobas^®^4800, HPV Test, Multiplex-RT-PCR, since 2020), and application of 5% acetic acid to the cervix represent the standard of care in our unit. This procedure is carried out for every woman who is referred with abnormal cytology. In addition, to examine the cervix and vagina more specifically, Lugol’s iodine is applied in some cases in order to visualize precancerous lesions in the vagina that were not visible beforehand.

The colposcopic findings are classified in accordance with the IFCPC into “normal” and “abnormal”, and are subdivided into “minor”, “major”, and “suspicious for invasion/cancer”. A distinction is also made between the findings as “nonspecific” and miscellaneous”. Normal findings include, for example, original squamous epithelium, columnar epithelium, or metaplastic squamous epithelium. Minor findings consist of fine punctuation and mosaic, thin acetowhite epithelium, and irregular and geographic borders. Sharp borders, an inner border sign, a ridge sign, dense acetowhite epithelium, a coarse mosaic pattern, and coarse punctuation represent typical major lesions. Atypical vessels, fragile vessels, irregular surface, exophytic lesions, necrosis, and ulceration are suspicious for invasion. Miscellaneous findings are represented by condylomas, polyps. or inflammation, and nonspecific lesions are leukoplakia or erosions [22]. For this study, the findings “normal”, “miscellaneous”, and “unspecific” were related to benign histology, as these findings are benign. Condylomas were regarded as benign histology, although some pathologists regard them as cervical intraepithelial neoplasia grade I (CIN I). If there is a major finding or a lesion that is suspicious for invasion, a colposcopy-directed biopsy has to be taken from the most suspicious part of the lesion, using biopsy forceps (Seidl Biopsy Forceps ER076R; Aesculap AG, Tuttlingen, Germany). In some patients with multifocal lesions, more than one biopsy is necessary. Decisions regarding surgical treatment are based on the colposcopic findings, types of TZ, the age of the women at the time of diagnosis, cytology results, HPV testing, and the histological findings. In some borderline cases (e.g., metaplasia or portio ectopy), a biopsy is taken in order to rule out neoplastic lesions, even when the examiners expect the results to be normal. 

During the period of this retrospective analysis, the team in the dysplasia unit consisted of 11 colposcopists with various degrees of clinical experience and training. They were divided into three groups: those with 0–5 years’ experience, those with 5–10 years’ experience, and those with over 10 years’ experience.

All data, including colposcopic findings, Pap smear and HPV test results, histological outcomes, number of biopsies, type of transformation zone, and epidemiological outcomes, were recorded prospectively in a database for further research.

In case of histologic findings differing between the biopsy and the result of the operation, the most severe histology was taken for comparison. The concordance rate is the percentage of patients with the same colposcopic findings and more severe histology: normal, miscellaneous, and unspecific colposcopic findings are equivalent to benign histology, minor findings to CIN I, and major findings to CIN II/CIN III/adenocarcinoma in situ (AIS), and suspicion of invasion is correlated to invasive cancer. Overdiagnosis was considered to be present if the colposcopic findings suspected a more advanced lesion than the histological result indicated, and underdiagnosis was present if the final histology showed a more advanced lesion than the colposcopic findings. 

Women with normal or minor changes were referred back for regular check-up examinations. If the biopsy revealed HSIL, a decision in favor of surgery was made. Depending on the size of the lesion, the (TZ), and the patient’s age, different types of conization were possible. Postmenopausal women, those with intracervical lesions, or those with a type 3 transformation zone underwent LLETZ. Women aged below 25 with a diagnosis of HSIL were selected for observation due to the strong chances of regression [25]. These women were scheduled for intensified follow-up examinations with cytology and colposcopy. Surgical excisional treatment was carried out if HSIL persisted or invasion was confirmed. In these cases, LEEP with laser coagulation of the periphery or laser conization was carried out. LEEP is also performed in women of reproductive age with TZ1 or extracervical lesions. Women of reproductive age with TZ3, intracervical lesions, or suspected microinvasion were treated with laser conization. Women with AIS were treated with laser conization due to the high risk of invasion. In women in whom the colposcopy-directed biopsy excluded invasion, the whole dysplasia was visible, and TZ1, ablative laser treatment was an option in order to minimize the damage done to the cervix. All of the operations were performed by experienced and highly qualified staff at Erlangen University Hospital [6].

### Statistical Analysis

To examine the agreement between the colposcopic findings and histology, we first calculated Spearman’s rank correlation. We then calculated overdiagnosis (findings worse than histology), concordance rate, and underdiagnosis (histology worse than findings) for the following four categories: (1) normal/miscellaneous/unspecific; (2) minor; (3) major; and (4) suspicious for cancer. Additionally, we formed the following two groups: group 1 (normal/miscellaneous/unspecific, minor) and group 2 (major, suspicious for cancer). For these groups, we calculated sensitivity, specificity, positive and negative predictive values, and the corresponding 95% confidence intervals. This analysis was performed for the total data and repeated for the subgroups specified by level of colposcopic experience (0–5, 5–10, >10 years), TZ (1, 2, 3), and age (<35 years and ≥35 years). Concordance between colposcopy and histology was evaluated in three logistic regression models with agreement (yes/no) as the dependent variable and colposcopic findings (reference: major lesions), TZ (reference: TZ1), and experience (reference: more than 10 years), respectively, as independent variables. All statistical analyses were conducted using the R V4.2.0 statistics program (R Core Team (2022). R: A language and environment for statistical computing. R Foundation for Statistical Computing, Vienna, Austria).

## 3. Results

A total of 11,086 colposcopies were performed during the period of this retrospective study. In 6308 cases, no histology during colposcopies or no surgery after colposcopy was necessary. This leaves a total of 4778 colposcopies with histology in 4001 women. Laser conization was performed in 601 cases, LLETZ in 627 cases, and LEEP in 736 women (see Figure 1). 

The mean age of the women included at the time of colposcopy was 36.8 years (standard deviation 10.8 years). The colposcopic findings were “normal/miscellaneous/unspecific” in 1008 cases, “minor” in 1044 cases, “major” in 2550 cases, and “suspicious for cancer” in 176 cases. The most severe histology was “benign” in 939 cases, “CIN I” in 1161 cases, “CIN II/CIN III” in 2480 cases, adenocarcinoma in situ (“AIS”) in 31 cases, and “carcinoma” in 167 cases. Normal/miscellaneous findings/unspecific findings had a concordance rate of 52.3% (491/939). Benign lesions were overdiagnosed in 47.7% of cases (448/939). The concordance rate for minor lesions was 48.3% (504/1161). The rates of underdiagnosis for CIN I/LSIL and CIN II/CIN III/HSIL were: 20.3% (236/1161) and 23.2% (576/2480), respectively. The rates of overdiagnosis for CIN I/LSIL; CIN II/CIN III/HSIL were: 36.3% (421/1161) and 2.5% (63/2480)), respectively. The concordance rate for major lesions was 74% (CIN II/HSIL; CIN III/HSIL, AIS) (1858/2511). The colposcopic findings of “suspicion of carcinoma” were correct in 51.1% of cases. Carcinoma was under diagnosed in 46.1% (77/167) (see Table 1 and Figure 2). The overall sensitivity, specificity, positive predictive value (PPV), and negative predictive value (NPV) for diagnostic differentiation of the two groups—normal/miscellaneous/unspecific/minor and major/suspicious for cancer—were: 77.8% (95% CI, 76.12% to 79.31%), 69.3% (95% CI, 67.31% to 71.30%), 76.4% (95% CI, 74.74% to 77.96%), and 71% (95% CI, 68.94% to 72.91%) (for the subgroups, see Table 2).

A total of 2028 women had a type 1 transformation zone (TZ1) at the point of colposcopy; 1399 and 1351 women had type 2 transformation zones (TZ2) or type 3 transformation zones (TZ3), respectively. The rate of concordance for major findings was 72.8% (862/1184) for TZ1, 76.6% (670/875) for TZ2, and 66.4% (326/491) for TZ3. CIN II/III/HSIL was correctly diagnosed as major findings in 77.8% for TZ1 (852/1095) and for TZ3 in 58% (322/555) (see Figure 3, Figure 4 and Figure 5). A logistic regression model showed that the difference between TZ1 and TZ2 was not statistically significant (*p* = 0.053), while the difference between TZ1 and TZ3 was significant (*p* = 0.009). The rate of concordance for major findings in the group of examiners with 0–5 years of experience was 65.9% (779/1182); in the group with 5–10 years’ experience it was 76% (308/405); and for examiners with more than 10 years of experience it was 80.1% (771/963). The difference between more than 10 years of experience and 0–5 years of experience was statistically significant (*p* < 0.001), but there was no significant difference between 5–10 years’ experience and more than 10 years of experience (*p* = 0.097). CIN II/III/HSIL was correctly diagnosed as representing a major lesion in 71.3% (769/1078) of cases by the examiners with 0–5 years of experience and in 80.6% of cases (767/952) by the group of examiners with more than 10 years of experience (see Figure 6, Figure 7 and Figure 8). The rate of concordance for carcinoma increased along with the examiners’ level of experience. For women under 35 years of age, the rate of concordance for major findings was 74.1% (1060/1431) and for women over the age of 35 it was 71.3% (798/1119) (see Figure 9 and Figure 10). 

## 4. Discussion

This retrospective, single-center study included 4778 colposcopies with histology. The most common colposcopic finding was “major”. In 167 cases, the examiners thought the lesion was suspicious for cancer. The concordance rate for major lesions was 74.6% and the rate of concordance for lesions “suspicious for cancer” was 51.1%. The rate of concordance was poorer for type 3 transformation zones. The rate of concordance increased along with the level of experience of the examiner.

In the literature, there is some controversy regarding under- and overestimation of colposcopic assessments. In some studies, cervical lesions are more often underestimated, while in other studies cervical lesions are more often overestimated [26,27]. The rates of overdiagnosis (19.6%, 938/4778) and underdiagnosis (18.8%, 897/4778) were fairly balanced. In a study by Ruan et al. including 1828 women, almost half of the HSILs and carcinomas were underestimated [27]. In the present study, 46.1% of the women had carcinomas underestimated, but 38.9% were diagnosed with at least HSIL, so that 85% of them received a surgical intervention after a colposcopic examination and no women were missed.

In a meta-analysis including 11 studies with 6370 participants, the sensitivity of the colposcopic impression ranged from 0.29 to 1.00 and the specificity from 0.12 to 0.88 [28]. The sensitivity in two other studies ranged from 56.29% to 64.72%, with a specificity range of 52.74% to 93.82%. The PPV and NPV rates ranged from 76.32% to 77.47% and 85.04% to 95.41%, respectively [10,28]. In the present study, the sensitivity and specificity values were 77.75% (95% CI, 76.12% to 79.31%) and 69.33% (95% CI, 67.31% to 71.30%), respectively. One possible explanation for this is that the women were diagnosed by highly trained members of staff who are specialized in diagnosing HSIL and carcinoma of the cervix. The PPV rates (95% CI, 76.38%; range 74.74% to 77.96%) were comparable to those reported in the literature and those for NPV (95% CI, 70.96%; range 68.94% to 72.91%) were lower.

The ability to carry out visual assessment of the cervix depends on the type of transformation zone. By definition, a complete visual assessment is not possible in a cervix with a type 3 transformation zone and intracervical lesions may be missed [24]. For TZ3, the concordance was lowest for the detection of CIN II/III/HSIL (58%. Minor lesions were also less likely to be detected in women with TZ3 (31.6%) (see Figure 3, Figure 4 and Figure 5). Ruan et al. stated that TZ3 was most common in women with normal and low-risk findings, while TZ1 and TZ2 were more frequently associated with HSIL and carcinoma. The authors recommend that special attention should be given to TZ1 and TZ2 [27]. We cannot confirm these data. In the present study, the majority of carcinomas were associated with TZ3. In stage IB and larger carcinomas, the TZ is most often infiltrated by tumor and the TZ can therefore not be assessed completely. Moreover, there were no differences between TZ2 and TZ3 with regard to the distribution of low-grade and high-grade lesions in the group of patients included here.

Colposcopy is a highly subjective examination method, and intra- and interobserver agreement can vary significantly even among expert colposcopists [14,29,30]. Surprisingly, in a retrospective review, Baum et al. report the highest rate of agreement for second-year residents (77%), with lower rates for third-year residents (75%) and fourth-year residents (73%) [14]. Interestingly, the rate of concordance was highest for nurse practitioners (92%). They performed a large number of colposcopies and saw their own patients during follow-up. They were therefore receiving good feedback on their own work [14]. For CIN II/III/HSIL, we also observed a slight decrease in the concordance rates for examiners with 0–5 years of experience and 5–10 years of experience (71.3% and 80.6%, respectively). For carcinoma, there was a continuous increase in concordance for the different levels of experience, at 42.6%, 56.7%, and 61.8%, respectively (see Figure 6, Figure 7 and Figure 8). In our certified dysplasia unit, every examiner performs 500–1000 colposcopies per year. In order to achieve the best rate of concordance, a colposcopist needs at least 5000–10,000 examinations and 10 years or more of experience. Since 2014, consulting practices for gynecological dysplasia are required to conduct at least 100 documented colposcopies per year, while gynecological dysplasia units require at least 300 per year in Germany. For colposcopists working in a dysplasia unit, each colposcopist needs at least 100 colposcopies. There is also a minimum of 30 dysplasias/carcinomas (consulting practice for gynecological dysplasia) and 150 dysplasias/carcinomas (dysplasia units) [9]. This certification system ensures that there is a certain level of quality in certified consulting practices and dysplasia units. Our data support this level of required cases and underline the need for qualified staff.

A new organized screening program was implemented in Germany in January 2020 [13]. Women between 20 and 34 years of age are continuing to have annual Pap smears, while women over the age of 34 receive a co-test comprising a Pap smear in combination with an hrHPV test every 3 years. All women aged 20–65 are invited for testing by their health insurance providers every 5 years [13,31,32,33,34,35,36]. We therefore investigated the differences between the two different age groups. The rate of concordance was higher for CIN I/LSIL, CIN II/III/HSIL, and AIS (46.2%, 80.5%, and 72.7%, respectively) for women under the age of 35 in comparison with the older age group (40.9%, 67.3%, and 45%, respectively). This might be because these women are of reproductive age and are more likely to have TZ1 or TZ2. In the younger age group, 278 of 2327 had TZ3 (11.9%) and in the older age group 1073 of 2451 had TZ3 (43.8%). This could explain the lower rate of concordance in the older age group. For carcinomas, the rate of concordance was higher in the older age group (34% vs. 62.4%) (see Figure 9 and Figure 10). One possible explanation might be the fact that the average age for cervical cancer in Germany is 55, so the examiners did not expect carcinoma; 60% of the carcinomas were suspected to be major lesions [37]. Careful examination is also necessary for younger women in order to rule out cervical cancer.

## 5. Strengths and Limitations

This study includes a large set of women who were seen in a certified dysplasia unit. This is a highly selected group of patients, the majority of whom were referred to the dysplasia unit due to suspicious cytology findings. The study is limited by the bias inherent in any retrospective study. All women with normal colposcopy examinations in whom biopsy was not performed were excluded. This eliminated potential false-negative colposcopies. The colposcopic findings are based on the examination and not on static images, as has been the case in other studies. The cytological and histological findings were analyzed in the same department, in some cases by the same examiner. The cytologists were aware of the colposcopic appearance and therefore knew whether there was a suspicious lesion. This may have influenced the results. No information was available regarding the HPV vaccination status of the women referred. The team treating the women is comparatively small, and the physicians are highly specialized in treating HSIL and cervical cancer.

## 6. Conclusions

Colposcopy is a cornerstone in the detection of cervical dysplasia and cancer. It is a feasible and effective method. The published data on concordance, sensitivity, specificity, PPV, and NPV rates for colposcopy are contradictory. This study reports data from a certified dysplasia unit with a large set of women. The rate of concordance was highest for detecting CIN III/HSIL. Almost half of the carcinomas were underdiagnosed. The rate of concordance was highest for TZ1 and TZ2 and was significantly lower for TZ3. To perform colposcopy at a professional level, colposcopists need at least 10 years’ experience. Young women must undergo careful examination to ensure that carcinomas are not missed.

## Figures and Tables

**Figure 1 diagnostics-12-02436-f001:**
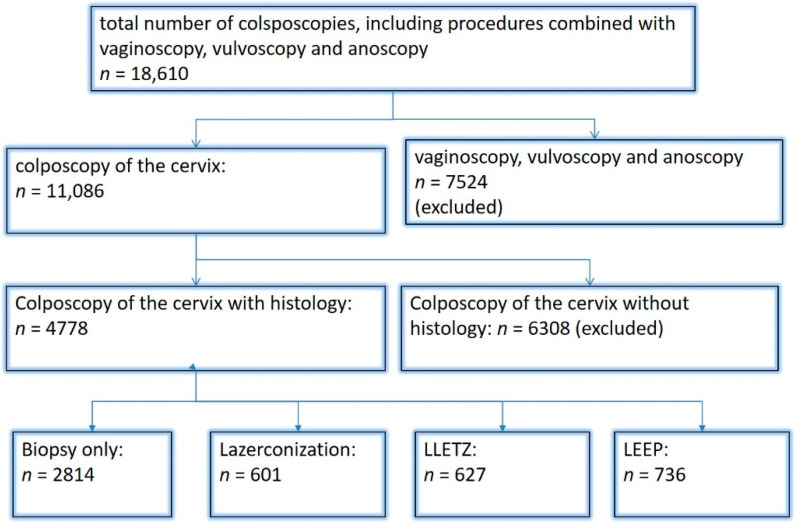
Inclusion and exclusion of patients.

**Figure 2 diagnostics-12-02436-f002:**
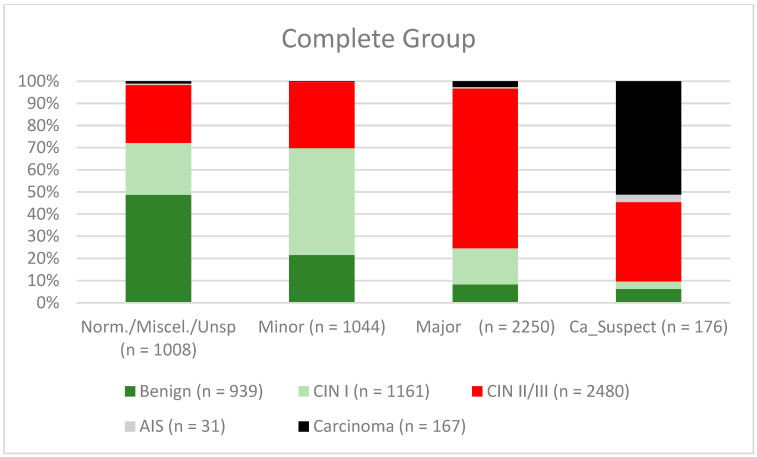
Findings vs. histology (all patients) (CIN, cervical intraepithelial neoplasia; AIS, adenocarcinoma in situ).

**Figure 3 diagnostics-12-02436-f003:**
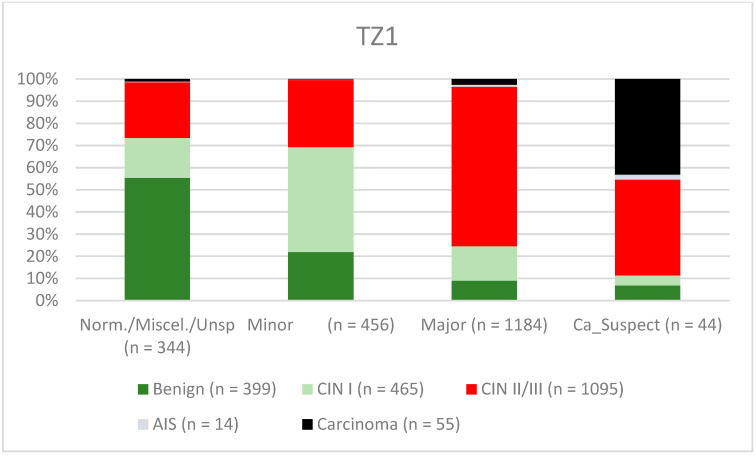
Findings vs. histology (type 1 transformation zone) (CIN, cervical intraepithelial neoplasia; AIS, adenocarcinoma in situ).

**Figure 4 diagnostics-12-02436-f004:**
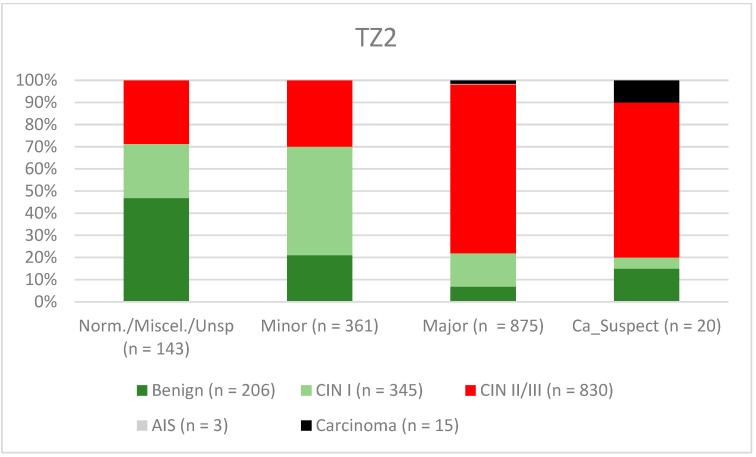
Findings vs. histology (type 2 transformation zone) (CIN, cervical intraepithelial neoplasia; AIS, adenocarcinoma in situ).

**Figure 5 diagnostics-12-02436-f005:**
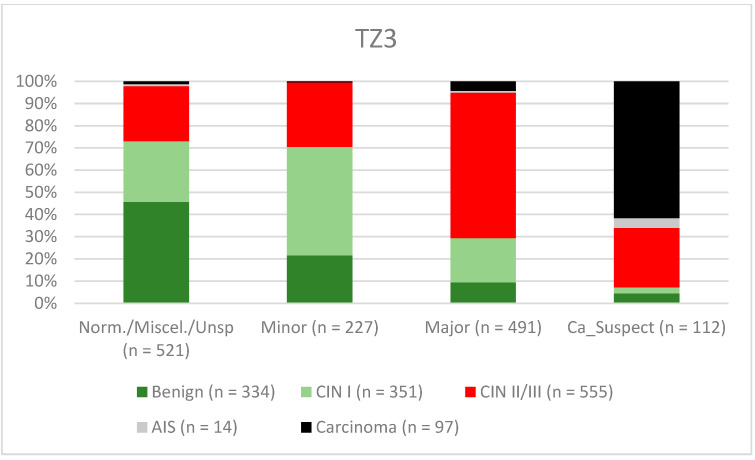
Findings vs. histology (type 3 transformation zone) (CIN, cervical intraepithelial neoplasia; AIS, adenocarcinoma in situ).

**Figure 6 diagnostics-12-02436-f006:**
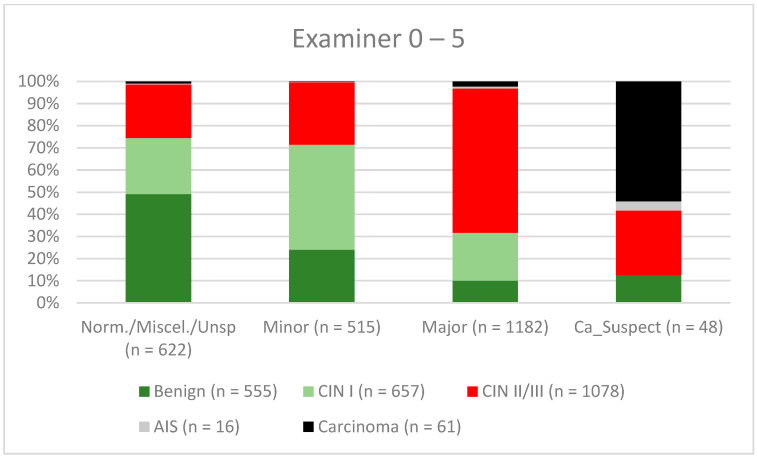
Findings vs. histology (examiners with 0–5 years’ experience) (CIN, cervical intraepithelial neoplasia; AIS, adenocarcinoma in situ).

**Figure 7 diagnostics-12-02436-f007:**
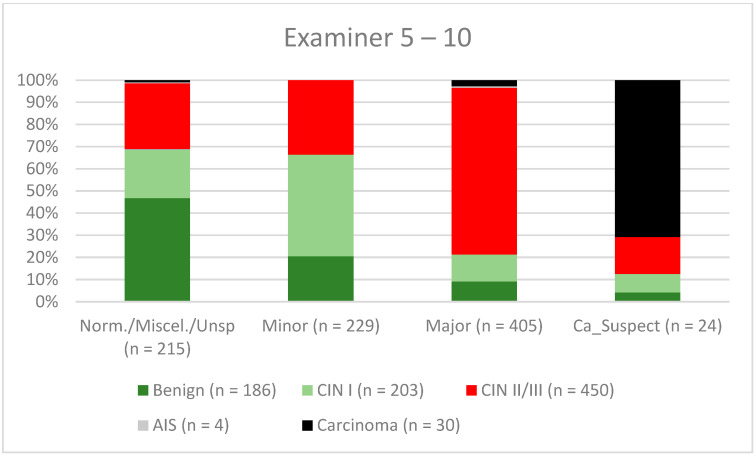
Findings vs. histology (examiners with 5–10 years’ experience) (CIN, cervical intraepithelial neoplasia; AIS, adenocarcinoma in situ).

**Figure 8 diagnostics-12-02436-f008:**
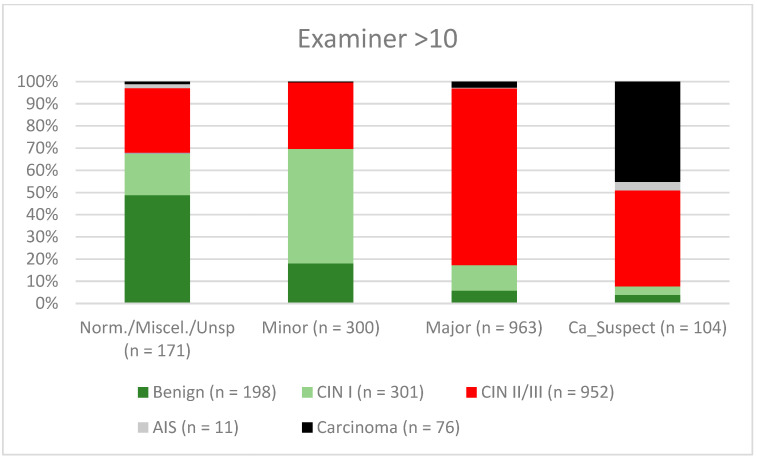
Findings vs. histology (examiners with >10 years’ experience) (CIN, cervical intraepithelial neoplasia; AIS, adenocarcinoma in situ).

**Figure 9 diagnostics-12-02436-f009:**
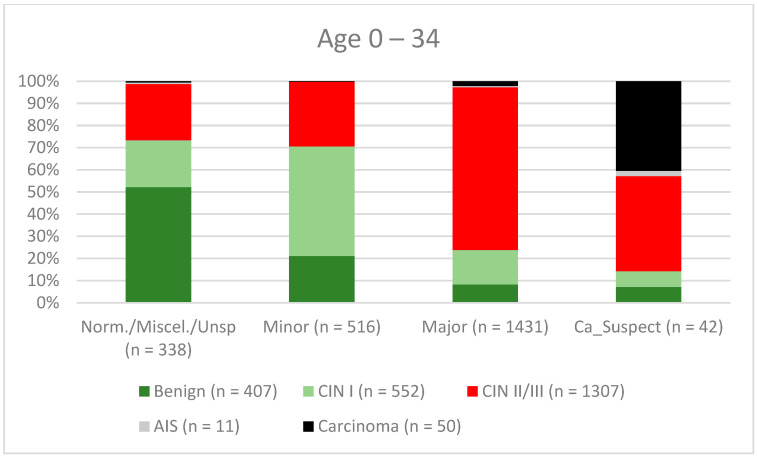
Findings vs. histology (patients aged 0–34 years) (CIN, cervical intraepithelial neoplasia; AIS, adenocarcinoma in situ).

**Figure 10 diagnostics-12-02436-f010:**
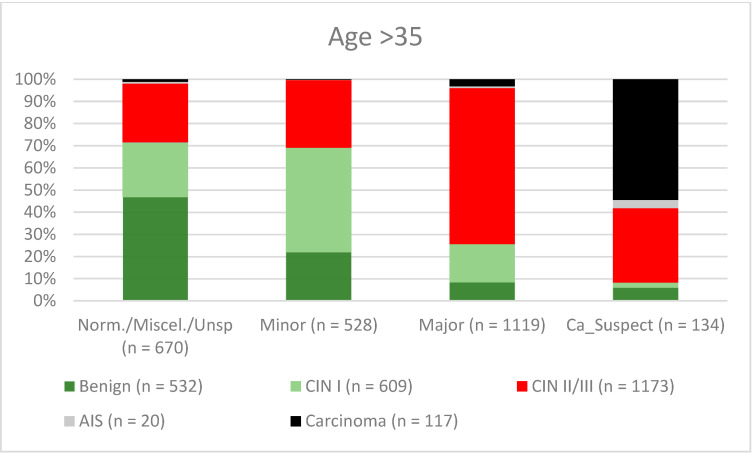
Findings vs. histology (patients aged > 35 years) (CIN, cervical intraepithelial neoplasia; AIS, adenocarcinoma in situ).

**Table 1 diagnostics-12-02436-t001:** Findings vs. histology (all patients).

Findings/Histology (*n* = 4778)	Benign (*n* = 939)	CIN I (*n* = 1161)	CIN II/III (*n* = 2480)	AIS (*n* = 31)	Carcinoma (*n* = 167)
	(*p* < 0.001)	(*p* < 0.001)	Reference		(*p* < 0.001)
Normal/miscellaneous/unspecific (*n* = 1008)	491 (52.3%)	236 (20.3%)	264 (10.6%)	7 (22.6%)	10 (6%)
Minor (*n* = 1044)	225 (24%)	504 (43.4%)	312 (12.6%)	1 (3.2%)	2 (1.2%)
Major (*n* = 2550)	212 (22.6%)	415 (35.7%)	1841 (74.2%)	17 (54.8%)	65 (38.9%)
Suspicious for cancer (*n* = 176)	11 (1.2%)	6 (0.5%)	63 (2.5%)	6 (19.4%)	90 (53.9%)

**Table 2 diagnostics-12-02436-t002:** Sensitivity, specificity, positive predictive value (PPV), and negative predictive value (NPV) for different subsets (TZ, transformation zone; CI, confidence interval).

	Sensitivity (95 CI)	Specificity (95 CI)	PPV (95 CI)	NPV (95 CI)
Complete group	77.8% (76.12–79.31%)	69.3% (67.31–71.30%)	76.4% (74.74–77.96%)	71.0% (68.94–72.91%)
TZ1	80.1% (77.66–82.33)	65.7% (62.47–68.9)	75.9% (73.4–78.26)	71.0% (67.72–74.12)
TZ2	82.4% (79.7–84.93)	64.4% (60.27–68.43)	78.1% (75.25–80.77)	70.4% (66.24–74.39)
TZ3	67.7% (64.02–71.26)	77.8% (74.51–80.87)	74.8% (71.13–78.21)	71.3% (67.87–74.48)
Examiner 0–5	73.5% (70.86–76.03)	68.6% (65.87–71.17)	69.0% (66.36–71.6)	73.1% (70.41–75.65)
Examiner 5–10	70.2% (65.89–74.24)	77.1% (72.62–81.2)	79.2% (75.05–82.95)	67.6% (62.99–71.9)
Examiner >10	86.0% (83.77–88.08)	65.1% (60.77–69.31)	83.7% (81.34–85.86)	69.2% (64.76–73.3)
Age 0–34	82.3% (80.18–84.3)	63.8% (60.68–66.86)	76.4% (74.19–78.59)	71.7% (68.51–74.66)
Age >35	73.0% (70.48–75.37)	74.0% (71.32–76.5)	76.3% (73.84–78.63)	70.5% (67.78–73.02)

## Data Availability

The data that support the findings of this study are available from the corresponding author upon reasonable request.

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
