# Peer review of "Concordance Rate of Colposcopy in Detecting Cervical Intraepithelial Lesions"

_diagnostics, 2022, doi:10.3390/diagnostics12102436_

Round 1
Reviewer 1 Report
Unfortunately, the paper contains several serious flaws. The terminology
used in the title, abstract, and throughout the text suggests that the
authors were evaluating the "diagnostic accuracy" of colposcopy. However, they rather assessed the concordance between the colposcopic impression obtained by colposcopists with different levels of experience and the final histology. As the declared primary aim of the study is "to estimate the rate of accuracy, sensitivity, specifity, PPV and NPV of colposcopy (see abstract), it is not understandable why the reader is only provided with information on the
"overall sensitivity, specificity, PPV and NPV" (three lines within the results) without any further differentiation, e.g. according to diagnosis, level of experience of the examiner, age of the patient, etc. The "diagnostic accuracy" of a test or diagnostic procedure is defined as the ratio of correct diagnoses (true positive + true negative) to the total number of subjects examined (true positive, true negative, false positive, false negative). However, the terminology used here (calculated by applying a test for correlation) seems to be used more as a synonym of "agreement" or "concordance".
The manuscript is not free from other terminological problems. For example, there is no "colposcopy of the anus" or "colposcopy of the vagina", the appropriate terms are respectively: anoscopy, vulvoscopy, vaginoscopy, etc. Certainly, colposcopy can be combined with these procedures. In addition to these major shortcomings, there are also many minor concerns, e.g. lack of information on incidence, mortality, etc. due to cervical cancer (globally and in the country where the study was performed), lacking information about authority responsible for certifying the center, or typographical errors (e.g. line 52).
Author Response
|
Dear Reviewer, Thank you very much for your time and effort. We have supplied you with a point-to-point reply and hope the changes made to the manuscript are now satisfactory. |
|
|
Unfortunately, the paper contains several serious flaws. The terminology used in the title, abstract, and throughout the text suggests that the authors were evaluating the "diagnostic accuracy" of colposcopy. However, they rather assessed the concordance between the colposcopic impression obtained by colposcopists with different levels of experience and the final histology. |
Thank you for this valuable comment. We have replaced the term “accuracy” with “concordance rate” throughout the whole manuscript. |
|
As the declared primary aim of the study is "to estimate the rate of accuracy, sensitivity, specifity, PPV and NPV of colposcopy (see abstract), it is not understandable why the reader is only provided with information on the "overall sensitivity, specificity, PPV and NPV" (three lines within the results) without any further differentiation, e.g. according to diagnosis, level of experience of the examiner, age of the patient, etc. The "diagnostic accuracy" of a test or diagnostic procedure is defined as the ratio of correct diagnoses (true positive + true negative) to the total number of subjects examined (true positive, true negative, false positive, false negative). However, the terminology used here (calculated by applying a test for correlation) seems to be used more as a synonym of "agreement" or "concordance". |
We have added a Table (Table 2) to show sensitivity, specificity, PPV and NPV rates for the different subsets relative to the diagnosis, level of experience of the examiner, and age of the patient to show the differences in the subsets. The term “accuracy” has been replaced with “concordance” throughout the whole manuscript. |
|
The manuscript is not free from other terminological problems. For example, there is no "colposcopy of the anus" or "colposcopy of the vagina", the appropriate terms are respectively: anoscopy, vulvoscopy, vaginoscopy, etc. Certainly, colposcopy can be combined with these procedures. |
Thank you for pointing out this important fact. We fully agree with your comment, but we are not aware that we used terms such as "colposcopy of the anus" or "colposcopy of the vagina". In this retrospective study, we focused on the colposcopy of the cervix. Cases including vulvoscopy or anoscopy were not included. |
|
In addition to these major shortcomings, there are also many minor concerns, e.g. lack of information on incidence, mortality, etc. due to cervical cancer (globally and in the country where the study was performed), lacking information about authority responsible for certifying the center, or typographical errors (e.g. line 52). |
We have added the rate of incidence and mortality rate globally and in Germany. Information about the authority responsible for certifying the dysplasia units and gynecological centers has also been added. The publication referring to the certification system has been cited. We have checked the whole manuscript for typographic errors. |

Reviewer 2 Report
Dear authors,
the manuscript is on an interesting topic. In my opinion, there are too many numbers and too few statistics in the article. After partial revision , the article will be suitable for publication.
Line 72:
30% of HSIL are reported to be missed by colposcopy alone [17].
Note:
Please, do not start the sentence with a number.
Lines 82-84:
The aim of the present study was to determine the accuracy of colposcopy in comparison with the final histological diagnosis obtained with a LLETZ, LEEP or laser conization.
Note:
The abbreviation LEEP (loop electrosurgical excision procedure) is not explained in the text. The abbreviation should be explained the first time it is used. The LLETZ abbreviation is explained on line 76.
Lines 131-132:
Decisions regarding surgical treatment are based on the cytology results, HPV testing, and
the histological findings.
NOTE:
Probably also on colposcopic findings, types of TZ, and on ages of a patient.
Lines 154-157:
Depending on the size of the lesion, the (TZ), and the patient’s age, different types of conization were possible. Postmenopausal women, those with intracervical lesions, or those with a type 3 transformation zone underwent LLETZ.
NOTE:
How were women with adenolesions treated? In which cases was laser conization performed? It should be specified and explained.
Lines 162-163:
In women in whom the colposcopy-directed biopsy excluded invasion, ablative treatment was an option in order to minimize the damage done to the cervix.
NOTE:
What does it mean? It means, for example, you performed ablative treatment on a woman with HSIL cytology and HR HPV positive testing and negative colposcopy-directed biopsy? Or it means that ablative treatment was performed on a woman with histologically proven CIN2/3 (=invasion was excluded by colposcopy-directed biopsy)? What kind of ablative treatment was used?
It should be specified and explained.
According to many experts, ablative treatment is not suitable for several reasons.
1) no histology is obtained
2) there is often TZ3 after ablative treatment
3) very superficial LEEP is as gentle as ablation
Lines 179-181 + Table 1
A total of 11.086 colposcopies were performed during the period of this retrospective study. In 6308 cases no histology during colposcopies or no operation after colposcopy was necessary. This leaves a total of 4778 colposcopies with histology in 4001 women.
NOTE:
According to Table 1, a total of 663 colposcopy-directed biopsies were performed in women with normal colposcopic findings. Why 663 women with normal colposcopic findings were biopsied and another 6308 were not? It must be explained.
It should be specified where the biopsy was performed (when the colposcopic finding was normal). Was only one biopsy performed? Or were multiple random biopsies performed?
Lines 189-190:
Normal and miscellaneous findings had an accuracy rate of 46.6% (309/663) and 53.1% (52/98), respectively.
+ Lines 125-127: Miscellaneous findings are represented by condylomata, polyps or inflammation and nonspecific lesions are leukoplakia or erosions [21].
NOTE:
Which histology result do you consider to be correct in colposcopic miscellaneous findings? E.g. condylomata are often evaluated as CIN 1 by a pathologist. If the colposcopic miscellaneous findings were assessed as benign, why do you list normal and miscellaneous colposcopic findings separately, when in both cases the correct colposcopy result are benign histological results ? In this case, these two parts should be connected.
Similarly, lines normal and miscellaneous and unspecific should be connected in Table 1. You correctly connected these three categories in the section Statistical Analysis (lines 169-171) = "We then calculated overdiagnosis (finding worse than histology), accuracy, and underdiagnosis (histology worse than finding) for the four categories 1) normal / miscallaneous /unspecific, 2) minor, 3) major, and 4) Cancer suspect."
Lines 197-200:
The overall sensitivity, specifity, positive predictive value (PPV) and negative predictice value (NPV): 77.75% (95% CI, 76.12% - 79.31%), 69.33% (95% CI, 67.31% - 71.30%), 76.38% (95% CI, 74.74% - 77.96%) and 70.96% (95% CI, 68.94% - 72.91%)
NOTE:
I do not understand it. The overall sensitivity, specifity, PPV and NPV of colposcopy for what?
For CIN 1-3 ?
For findings which you call major lesions (CIN II/HSIL; CIN III/HSIL, AIS) ?
For CIN 2+ (CIN 2, CIN 3 , AIS, ca) ?
For carcinoma ?
For group 1 or 2 defined in Statistical Analysis part ?
You write in the Statistical Analysis section, that you calculated sensitivity, specificity, positive and negative predictive values for both groups.
This must be clearly explained.
Lines 201-203:
Table 1
Note: Table 1 should be modified. The purpose of this work is whether the colposcopic findings correspond to the histological results. Therefore, it does not make sense to divide the results of HSIL histology into CIN 2 and CIN 3 in the case of colposcopic findings of major lesions.
Lines normal and miscellaneous and unspecific should be connected.
Columns CIN 2 and CIN 3 should be connected. You can leave AIS column separately, but you shloud create column major lesions (CIN 2/3/AIS), which corresponds to colposcopic major findings.
Table 1
Note: It would be interesting to statistically evaluate (and add to Table 1) the accuracy of colposcopy among categories 1) normal / miscallaneous /unspecific, 2) minor, 3) major, and 4) Cancer suspect. (line 171) in relation to major lesions accurancy using p-value.
Line 216
2028 women had a transformation zone type 1 (TZ1) at the point of colposcopy.
NOTE:
Please, do not begin the sentence with a number. Use e.g. A total of 2028 ...
Lines 217-219:
The rate of accuracy for major findings was 72.8% (862/1184) for TZ1, 76.6% (670/875) for TZ2 and 66.4% (326/491) for TZ3.
NOTE:
Surprisingly, the numbers are very similar. Add a statistic p-value between results in relation to accuracy for TZ1.
Lines 220-223:
The rate of accuracy for major findings in the group of 0-5 years of experienced examiners was 65.9% (779/1182), in the group of 5-10 years of experienced examiner 76% (308/405) and for examiners with more than 10 years of experience 80.1% (771/963).
NOTE:
It would be good to know if the results are statistically different. Add a p-value in relation to accuracy rate for examiners with more than 10 years of experience.
Author Response
|
Dear Reviewer, Thank you very much for your time and effort. We have supplied you with a point-to-point reply and hope the changes made to the manuscript are now satisfactory. |
|
|
Line 72: 30% of HSIL are reported to be missed by colposcopy alone [17].
Note: Please, do not start the sentence with a number. |
We have changed the sentence to “A total of 30% […]” |
|
Lines 82-84: The aim of the present study was to determine the accuracy of colposcopy in comparison with the final histological diagnosis obtained with a LLETZ, LEEP or laser conization.
Note: The abbreviation LEEP (loop electrosurgical excision procedure) is not explained in the text. The abbreviation should be explained the first time it is used. The LLETZ abbreviation is explained on line 76. |
This is a very important comment. We have explained LEEP in line 87–88 as well. |
|
Lines 131-132: Decisions regarding surgical treatment are based on the cytology results, HPV testing, and the histological findings.
NOTE: Probably also on colposcopic findings, types of TZ, and on ages of a patient. |
Of course “colposcopic findings, types of TZ, and the patient’s age” influence the decision regarding the type of treatment. Nevertheless, the cytology results and histological findings are most important. We have added the other factors as you suggest. |
|
Lines 154-157: Depending on the size of the lesion, the (TZ), and the patient’s age, different types of conization were possible. Postmenopausal women, those with intracervical lesions, or those with a type 3 transformation zone underwent LLETZ.
NOTE: How were women with adenolesions treated? In which cases was laser conization performed? It should be specified and explained. |
We are grateful for this comment and have added the different surgical treatments used in our cervical dysplasia unit. LLETZ is performed in postmenopausal women, who won’t deliver any children, because the cone is much larger in LLETZ than in laser conization and the risk for preterm delivery is therefore much higher after LLETZ than after laser conization. LEEP is the surgical option that is least invasive and is therefore used frequently in women of reproductive age. Lazer conization is used in women with TZ3 or with findings suspicious for microinvasion in women of reproductive age. Women with adenocarcinoma in situ (ACIS) were treated with laser conization due to the high risk of invasion. In women with ACIS who are not intending to have children, a hysterectomy is performed. |
|
Lines 162-163:
In women in whom the colposcopy-directed biopsy excluded invasion, ablative treatment was an option in order to minimize the damage done to the cervix.
NOTE: What does it mean? It means, for example, you performed ablative treatment on a woman with HSIL cytology and HR HPV positive testing and negative colposcopy-directed biopsy? Or it means that ablative treatment was performed on a woman with histologically proven CIN2/3 (=invasion was excluded by colposcopy-directed biopsy)? What kind of ablative treatment was used? It should be specified and explained.
According to many experts, ablative treatment is not suitable for several reasons.
1) no histology is obtained 2) there is often TZ3 after ablative treatment 3) very superficial LEEP is as gentle as ablation |
One treatment option in women of reproductive age in whom cytology and biopsy excluded invasion and with TZ1 and TZ2 is ablative laser treatment. Laser ablation is a very gentle treatment for women intending to have children, in order to prevent preterm delivery. After conization, the rate of TZ3 is also very high. |
|
Lines 179-181 + Table 1
A total of 11.086 colposcopies were performed during the period of this retrospective study. In 6308 cases no histology during colposcopies or no operation after colposcopy was necessary. This leaves a total of 4778 colposcopies with histology in 4001 women.
NOTE: According to Table 1, a total of 663 colposcopy-directed biopsies were performed in women with normal colposcopic findings. Why 663 women with normal colposcopic findings were biopsied and another 6308 were not? It must be explained.
It should be specified where the biopsy was performed (when the colposcopic finding was normal). Was only one biopsy performed? Or were multiple random biopsies performed? |
This is a very important comment. Diagnosis of the cervix can be very challenging at times, especially for inexperienced examiners. For example, in women who are thought to have metaplasia, the biopsy reveals dysplasia in some cases, or in women with portio ectopy a biopsy is frequently taken to rule out CIN or microinvasion. Patients who have normal findings but who have a biopsy taken represent mainly borderline cases. As you point out, a biopsy was taken in only approximately 10% of patients with normal findings. It is our philosophy to take biopsies generously in borderline cases in order to rule out neoplastic lesions, even when the examiners expect the results to be normal. Unfortunately, we have not recorded how many biopsies were taken. We are aware that detection rate is higher when more than one biopsy is taken. |
|
Lines 189-190:
Normal and miscellaneous findings had an accuracy rate of 46.6% (309/663) and 53.1% (52/98), respectively.
+ Lines 125-127: Miscellaneous findings are represented by condylomata, polyps or inflammation and nonspecific lesions are leukoplakia or erosions [21].
NOTE: Which histology result do you consider to be correct in colposcopic miscellaneous findings? E.g. condylomata are often evaluated as CIN 1 by a pathologist. If the colposcopic miscellaneous findings were assessed as benign, why do you list normal and miscellaneous colposcopic findings separately, when in both cases the correct colposcopy result are benign histological results ? In this case, these two parts should be connected. Similarly, lines normal and miscellaneous and unspecific should be connected in Table 1. You correctly connected these three categories in the section Statistical Analysis (lines 169-171) = "We then calculated overdiagnosis (finding worse than histology), accuracy, and underdiagnosis (histology worse than finding) for the four categories 1) normal / miscallaneous /unspecific, 2) minor, 3) major, and 4) Cancer suspect." |
This is a very important comment. In the Rio 2011 classification, the findings normal, miscellaneous and unspecific are stated separately, as you know. We therefore found it interesting to investigate whether there is a difference in accuracy between the different findings. Nevertheless, we totally agree that the findings all correlate with benign histology. That is why we initially formed the four groups as described in the statistics section. We have now altered the whole manuscript as suggested by you and formed one group for Normal/miscellaneous/unspecific. We have altered the methods, results and discussions sections accordingly. Condylomas are classed as benign histology in this context. |
|
Lines 197-200:
The overall sensitivity, specifity, positive predictive value (PPV) and negative predictice value (NPV): 77.75% (95% CI, 76.12% - 79.31%), 69.33% (95% CI, 67.31% - 71.30%), 76.38% (95% CI, 74.74% - 77.96%) and 70.96% (95% CI, 68.94% - 72.91%)
NOTE:
I do not understand it. The overall sensitivity, specifity, PPV and NPV of colposcopy for what?
For CIN 1-3 ? For findings which you call major lesions (CIN II/HSIL; CIN III/HSIL, AIS) ? For CIN 2+ (CIN 2, CIN 3 , AIS, ca) ? For carcinoma?
For group 1 or 2 defined in Statistical Analysis part ? You write in the Statistical Analysis section, that you calculated sensitivity, specificity, positive and negative predictive values for both groups.
This must be clearly explained. |
We have made the analysis more specific in the manuscript: The overall sensitivity, specificity, positive predictive value (PPV) and negative predictive value (NPV) for diagnostic differentiation between the two groups normal / miscellaneous / unspecific / minor (normal group) and major / suspicious for cancer (diseased group) are: […] |
|
Lines 201-203:
Table 1
Note: Table 1 should be modified. The purpose of this work is whether the colposcopic findings correspond to the histological results. Therefore, it does not make sense to divide the results of HSIL histology into CIN 2 and CIN 3 in the case of colposcopic findings of major lesions.
Lines normal and miscellaneous and unspecific should be connected.
Columns CIN 2 and CIN 3 should be connected. You can leave AIS column separately, but you shloud create column major lesions (CIN 2/3/AIS), which corresponds to colposcopic major findings.
Table 1
Note: It would be interesting to statistically evaluate (and add to Table 1) the accuracy of colposcopy among categories 1) normal / miscallaneous /unspecific, 2) minor, 3) major, and 4) Cancer suspect. (line 171) in relation to major lesions accurancy using p-value |
We have connected the lines and columns as you suggest and have also done this for the different figures.
We have added the statistical analysis suggested by the reviewer. In the "Statistical Analysis" section we have added: Concordance between colposcopy and histology was evaluated in three logistic regression models with agreement (yes/no) as the dependent variable and colposcopic findings (reference: major lesions), TZ (reference: TZ1), and experience (reference: > 10 years), respectively, as independent variables. We have added P values for the comparison between categories 1) and 3), 2) and 3), and 3) and 4) to Table 1.
|
|
Line 216
2028 women had a transformation zone type 1 (TZ1) at the point of colposcopy.
NOTE:
Please, do not begin the sentence with a number. Use e.g. A total of 2028 ... |
We have changed this as suggested by you. |
|
Lines 217-219:
The rate of accuracy for major findings was 72.8% (862/1184) for TZ1, 76.6% (670/875) for TZ2 and 66.4% (326/491) for TZ3.
NOTE:
Surprisingly, the numbers are very similar. Add a statistic p-value between results in relation to accuracy for TZ1. |
We have added the statistical analysis suggested (see comment above). We have added the sentence: A logistic regression model showed that the difference between TZ1 and TZ2 was not statistically significant (P=0.053), while the difference between TZ1 and TZ3 was (P=0.009). |
|
Lines 220-223:
The rate of accuracy for major findings in the group of 0-5 years of experienced examiners was 65.9% (779/1182), in the group of 5-10 years of experienced examiner 76% (308/405) and for examiners with more than 10 years of experience 80.1% (771/963).
NOTE:
It would be good to know if the results are statistically different. Add a p-value in relation to accuracy rate for examiners with more than 10 years of experience. |
We have added the suggested statistical analysis (see comment above).
We have added the sentence: Here, the difference between more than 10 years of experience and 0-5 years of experience was statistically significant (P<0.001), while there was no significant difference between 5-10 years and more than 10 years of experience (P=0.097). |

Round 2
Reviewer 1 Report
The authors have made a careful revision. In fact, the manuscript has improved greatly. A small issue still remained unaddressed, namely the misleading term "total colposcopies including cervix, vagina, vulva, anus" (Figure 1, first box from top). I suggest "total number of colsposcopies, including procedures combined with anoscopy, vluvoscopy, or vaginascopy", or similar. In summary, I appreciate the efforts of the authors, congratulate them on their work and recommend publication after correcting Figure 1.
Author Response
Dear Reviewer,
Thank you very much again for your time and effort. We have corrected figure1 as suggested by you. The termy colposcopy of the vagina, vulva and anus were replaced by anoscopy, vulvoscopy, or vaginoscopy.
Best Regards
F. Stübs

Reviewer 2 Report
Dear authors,
after minor adjustments, the article will be suitable for publication.
Line 152
abbreviation CIN is not explained
Line 159
caes, correct cases
Line 177
abbreviation AIS is not explained
Line 194
Why do you use abbreviation ACIS on line 194 and AIS on line 177 ?
Author Response
Dear Reviewer,
Thank you very much again for your time and effort. We have corrected the manuscript as suggested by you.
Line 152
abbreviation CIN is not explained
cervical intraepithelial neoplasia = CIN. This was included into the manuscript.
Line 159
caes, correct cases
We corrected it.
Line 177
abbreviation AIS is not explained
Adenocarcinoma in situ = AIS. This was included into the manuscript.
Line 194
Why do you use abbreviation ACIS on line 194 and AIS on line 177 ?
We changed it in line 194 into AIS so it is equal throughout the whole manuscript.
Best Regards
F. Stübs
